# Graph Contrastive Learning with Cohesive Subgraph Awareness

## ABSTRACT

Graph contrastive learning (GCL) has emerged as a state-of-the-art strategy for learning representations of diverse graphs including social and biomedical networks. GCL widely uses stochastic graph topology augmentation, such as uniform node removal, to generate augmented graphs. However, such stochastic augmentations may severely damage the intrinsic properties of a graph and deteriorate the following representation learning process. Specifically, cohesive topological properties (e.g., $k$-core and $k$-truss) indicate strong and critical connections among multiple nodes; randomly removing nodes from a cohesive subgraph may remarkably alter the graph properties. In contrast, we argue that incorporating an awareness of cohesive subgraphs during the graph augmentation and learning processes has the potential to enhance GCL performance. To this end, we propose a novel unified framework called *CTAug*, to seamlessly integrate cohesion awareness into various existing GCL mechanisms. In particular, *CTAug* comprises two specialized modules: *topology augmentation enhancement* and *graph learning enhancement*. The former module generates augmented graphs that carefully preserve cohesion properties, while the latter module bolsters the graph encoder's ability to discern subgraph patterns. Theoretical analysis shows that *CTAug* can strictly improve existing GCL mechanisms. Empirical experiments verify that *CTAug* can achieve state-of-the-art performance for both graph and node representation learning, especially for graphs with high degrees.[1]

## CCS CONCEPTS

• **Information systems** → **Social networks**; • **Computing methodologies** → *Unsupervised learning*.

## KEYWORDS

social networks, graph contrastive learning, self-supervised learning, cohesive subgraph

**ACM Reference Format:**
Anonymous Author(s). 2018. Graph Contrastive Learning with Cohesive Subgraph Awareness. In *Proceedings of Make sure to enter the correct conference title from your rights confirmation emai (Conference acronym 'XX)*. ACM, New York, NY, USA, 15 pages. https://doi.org/XXXXXXX.XXXXXXX

[1]Codes: https://anonymous.4open.science/r/CTAug-BFD6

## 1 INTRODUCTION

Graph contrastive learning (GCL) has become a promising self-supervised learning paradigm to learn graph and node embeddings for various applications, such as social network analysis and web graph mining [27, 48, 58, 60]. The idea of GCL is maximizing the representation consistency between different augmented views from the same original graph [54], in order to learn an effective graph neural network encoder. Hence, the augmentation strategies for view generation play a vital role in GCL. In general, there are two augmentation types, i.e., *topology* and *feature* [58]. In this paper, we focus on topology augmentation, as it can be applied to either attributed or unattributed graphs.

Common topology augmentation strategies include node dropping, edge removal, subgraph sampling, etc. [58]. Existing methods mainly follow a stochastic manner to conduct topology augmentation [54, 59]. Some methods adopt total randomized augmentation operations, like removing nodes or edges with an equivalent probability. Concerning that nodes and edges usually hold diverse levels of importance in a graph, some other methods argue that a better augmentation strategy should more likely retain the more important components of the original graph. Otherwise, randomly deleting important edges/nodes may cause the augmented views to vary far away from the original graph, thus degrading the learned graph/node embedding. Recently, some pioneering work starts leveraging the intrinsic properties of a graph or domain knowledge to guide the graph augmentation of GCL [41, 46, 56, 60]. For example, *GCA* [60] introduces edge centrality into topology augmentation, so that important edges are likely to be kept after augmentation. Nevertheless, there remain some important research questions.

**1. Property Enrichment**. Very limited types of properties about graphs have been used to determine important components of a graph and enhance graph augmentation for effective GCL. However, a basket of individual-level (i.e., node/edge) and structure-level intrinsic graph properties have been defined to distinguish the importance of elements in real-life social graphs; such properties have also been used to improve a variety of applications [16, 47]. Can we enrich the topology augmentation with more essential graph properties to improve GCL?

**2. Unified Framework**. Most existing studies focus on designing a concrete GCL mechanism for representation learning. However, as topology augmentation is a widely adopted step in various mechanisms [58], can we develop a unified framework to incorporate graph properties into all of these GCL mechanisms and benefit graph representation learning?

**3. Expressive Network.** Most existing GCL methods [15, 54] use standard Graph Neural Networks (GNNs) such as GCN [20] and GIN [50] as GNN encoder. However, prior research has indicated that GNNs have limited expressive power and encounter difficulties in capturing subgraph properties [9]. Can we engineer a more expressive graph encoder that can effectively capture subgraph information from the original graph?

This research serves as a pioneering effort to address the above research questions. Firstly, we propose to introduce *cohesive* subgraphs to guide topology augmentations, which provide a novel *structural-level* view of a graph's properties for graph augmentations. In general, cohesive subgraphs are densely connected subsets of important nodes in a graph. A broad of cohesive subgraphs with different specific semantic definitions, including $k$-clique [28], $k$-core [4, 36], and $k$-truss [10], have been investigated in the graph theory literature and regarded as critical structures of graphs in a spectrum of domains such as social networks and World Wide Web [12, 18, 23]. Therefore, the basic idea of cohesion-guided augmentation is preserving *cohesive* subgraphs of a graph in its augmented views. While the existing literature primarily depends on node-level graph properties or domain knowledge, *cohesive* subgraphs could provide an effective complement to the properties studied in the literature (e.g., centrality [60]).

Moreover, we propose a unified topology augmentation framework *CTAug* to ensure that the cohesion-guided augmentation idea could be flexibly adapted into a variety of graph augmentation methods. While the predominant augmentation methods fall into either the *probabilistic* or *deterministic* categories, *CTAug* customizes two distinct strategies to cater to these methods. In the realm of *probabilistic* augmentation-based GCL methods for graph-level representation learning [53, 54], diverse augmented views are generated in a stochastic manner. *CTAug* refines perturbation probability to create augmented views that specifically retain the most cohesive subgraphs from the original graph. Besides, *deterministic* methods typically follow a well-defined procedure to produce a single fixed augmented view [15]. In this context, *CTAug* preserves the established procedure of a particular deterministic method but modifies the original graph by increasing the weights of nodes and edges within cohesive subgraphs. With this design, the augmented graphs are supposed to better preserve cohesive subgraphs of the original graph.

However, existing research has pointed out that plain GNNs are hard to capture subgraph properties [9], which results in the loss of cohesive subgraph information during the graph representation learning process. To address this, inspired by [7], we then propose an *original-graph-oriented graph substructure network* (O-GSN) to enhance GNNs' power to aware graph cohesive substructures efficiently when encoding graphs. Besides, We also extend *CTAug* for GCL methods of node-level representation learning [60].

In summary, this paper makes the following contributions.

1. To the best of our knowledge, this is one of the first studies to incorporate cohesion properties into GCL. Considering cohesion as a type of *graph intrinsic knowledge* [60], this research sheds light on incorporating knowledge into self-supervised graph learning paradigms.

2. We propose *CTAug*, a unified framework that can consider multiple types of cohesion properties in various GCL mechanisms during topology augmentation and graph learning processes. Theoretical analysis on the superiority of *CTAug* over conventional GCL methods is provided.

3. Extensive experiments on real-life datasets validate that *CTAug* can significantly improve existing GCL mechanisms, such as GraphCL [54], JOAO [53], MVGRL [15], and GCA [60], especially for graphs with high degrees.

## 2 BACKGROUND AND RELATED WORK

### 2.1 Cohesive Subgraph

In literature, various cohesive subgraphs have been studied in graphs [5, 28]. In this paper, we focus on two widely-studied ones, $k$-core [36] and $k$-truss [10], as they both have efficient computation algorithms in polynomial time [4, 45].

$k$**-core** is a maximal subgraph in which every node has at least $k$ links to the other nodes [36]. As an extension to $k$-core, $k$-shell is a subgraph including the nodes that are in $k$-core but not in $(k + 1)$-core. Finding $k$-core and $k$-shell is efficient as the time complexity is linear to the edge number [4]. Analyzing such a subgraph can provide rich information for applications in various social network applications [23], such as user influence [2, 8, 21, 55] and community detection [13, 31]. For instance, researchers find that $k$-core plays an important role in analyzing coauthor social networks [13]. Specifically, it is easy to know that a paper with $(k + 1)$ authors can lead to a $k$-core subgraph in a coauthor network (i.e., every author is linked to the other $k$ authors as they have the paper collaboration) [13]; then, as different research topics usually hold diverse collaboration styles (some topics need a large research team, i.e., many coauthors, but some do not), $k$-core could be an effective indicator to infer the research domain of a given coauthor network. In addition to social networks, $k$-core is verified as an important property for bioinformatics [3], digital library text mining [34], airline networks [49], etc.

$k$**-truss** is the largest subgraph in which every edge is in at least $(k - 2)$ triangles of the subgraph [10]. Triangle is the fundamental building element for networks and can indicate the stability of the social network topology, as quantified by the *clustering coefficient* [47]. Triangle also reveals the transitivity in the link formation of networks [16]. This provides an effective indicator for link prediction in social networks [17]. Besides, researchers point out that the triangles in the hyperlink-based web graph reveal the topic distribution over the World Wide Web [12]. As a common way to measure triangles in subgraphs, $k$-truss has thus attracted much research interest in network analysis [1, 18].

### 2.2 Topology Augmentation in GCL

Topology augmentation is widely adopted in GCL [58]. There are two main types of topology augmentation strategies, i.e., *probabilistic* and *deterministic*.

Most topology augmentation strategies in GCL are probabilistic, such as stochastic node dropping, edge perturbation, and subgraph sampling [53, 54, 59, 60]. More specifically, most traditional probabilistic strategies are purely randomized. For instance, the probability of the topology augmentation operations is set to *uniform* over all the nodes and edges in *GraphCL* [54, 59]. More recently, some studies have tried to adaptively learn *non-uniform* probabilities. One stream of work uses intrinsic knowledge to guide topology augmentation, such as centrality [60] and motif [56]. Another stream of work uses a data-driven way to automatically adjust the probabilities [24, 39, 52]. Our work follows the first stream by introducing the *cohesion* property into topology augmentation.

Some studies adopt a deterministic strategy in topology augmentation — given an original graph, the augmented view is fixed. The representative strategies are *diffusion*-based augmentations

[15, 58]. Conceptually, the diffusion operation would add edges to the original graph. Different from the probabilistic edge adding [54], the diffusion process is computed in a deterministic and analytic manner, e.g., following the Personalized PageRank [15] or Markov Chain processes [57].

Recently, there has been emerging research in topology augmentation from the *spectral* domain. [26] suggests that GCL primarily encodes low-frequency information, whereas [25] focuses on maximizing spectral changes during augmentation. However, it's notable that spectral features are computed based on the entire graph Laplacian matrix, and using them to guide augmentation typically requires intricate transformations of the corresponding graph topology. This may lead to substantial computational costs and a lack of intuitive interpretation.

Unlike most prior work, our study does not aim to provide a concrete GCL mechanism. Instead, our goal is to improve existing GCL mechanisms by incorporating the concept of cohesion in topology augmentation. Recently, a review of existing GCL methods [41] highlighted that injecting domain knowledge of graphs in GCL may lead to better performance. Our work aligns with this direction and demonstrates the effectiveness of considering cohesion as a knowledge factor in GCL. Besides, our method are primarily concerned with graph augmentation within the spatial domain, leading to more conciseness and explainability.

## 3 THE *CTAUG* FRAMEWORK

GCL aims to learn graph representations by maximizing agreement between similar graphs and minimizing agreement between dissimilar graphs. The basic loss function for a pair of graphs $\mathcal{G}_1$ and $\mathcal{G}_2$ with representations $z_1$ and $z_2$ is [54]:

$$L = -\log \frac{\exp(sim(z_1, z_2)/\tau)}{\sum_{i,j} \exp(sim(z_i, z_j)/\tau)} \tag{1}$$

where $\tau$ is a temperature parameter, $sim$ is cosine similarity with $sim(z_i, z_j) = z_i^T z_j / \|z_i\| \|z_j\|$. For similar graph pairs $(\mathcal{G}_1, \mathcal{G}_2)$ augmented from the same graph (e.g., dropping nodes or edges with a probability $p_{dr}$), $z_1$ and $z_2$ should be close, so the numerator is large and the loss is small. For dissimilar pairs augmented from different original graphs, the denominator becomes large and the loss increases.

As shown in Fig. 1, *CTAug* consists of two modules that respectively enhance the topology augmentation and graph learning steps in GCL methods. The first module modifies the augmentation process to generate augmented graphs that preserve the cohesion properties of the original graph. The second module improves the GNN encoder to produce graph representations that better capture the original graph's cohesion properties. By jointly applying these two modules, *CTAug* aims to highlight the cohesion properties of graphs throughout the GCL pipeline.

### 3.1 Topology Augmentation Enhancement

*3.1.1 **Probabilistic Topology Augmentation**.* In general, the probabilistic topology augmentation methods may generate a variety of augmented graphs with probabilistic network manipulation operations [54]. *CTAug* intends to make probabilistic augmented graphs retain more cohesive components of the original graph.

A straightforward method is firstly generating multiple candidate augmented graphs and selecting the one most similar to the original graph regarding a particular cohesion property. However, generating multiple augmented graphs and computing their cohesive subgraphs is time-consuming. To address this, we propose to refine the probability of augmentation operations to make that nodes and edges in cohesive subgraphs likely retain in augmented graphs. Then, we need to generate only one augmented graph, while it would tend to keep certain cohesion properties as the original graph.

Specifically, we reduce the probability of node-dropping or edge-dropping operations on the cohesive subgraphs of the original graph. With this idea in mind, *CTAug* multiplies the original dropping probability $p_{dr}$ relevant to the nodes and edges in the cohesive subgraphs by a decay factor $\epsilon \in (0, 1]$, leading to a newly-refined dropping probability,

$$p'_{dr} = (1 - \epsilon) \cdot p_{dr} \tag{2}$$

For instance, suppose that the original node dropping probability $p_{dr}$ is uniformly set as 0.2 [54]. Then, by setting $\epsilon = 0.5$, the dropping probabilities for the nodes in a cohesive subgraph will be reduced to $0.2 \times 0.5 = 0.1$. With the newly-refined node-dropping and edge-dropping probability, we can continue running existing GCL mechanisms without the need for making other modifications.[2]

More specifically, for a certain cohesion property, e.g., $k$-core, the parameter $k$ can be varied, and then various subgraphs are extracted from an original graph. To consider cohesive subgraphs of varying $k$, first, given the original graph $\mathcal{G}$, we range $k$ from $k_{\min}$ to $k_{\max}$ and thus extract a set of $k$-core subgraphs,

$$\mathbb{S} = \{\mathcal{S}_{core}^k | k = k_{\min}, k_{\min} + 1, ..., k_{\max}\} \tag{3}$$

where $k_{\max}$ is the order of the main core (the core with the largest order) of the original graph, $k_{\min}$ can be set to $\max\{k_{\max} - 2, 1\}$ for $k$-core and $\max\{k_{\max} - 2, 2\}$ for $k$-truss. For a vertex $v_i$, we count how many times it appears in the set of subgraphs $\mathbb{S}$ to calculate its importance weight $w_v$.

$$w_v(v_i) = \sum_{\mathcal{S} \in \mathbb{S}} \mathbf{1}_{v_i \in vertex(\mathcal{S})} \tag{4}$$

where $\mathbf{1}_{v_i \in vertex(\mathcal{S})}$ is an indicator function to output whether $v_i$ is in the vertex set of $\mathcal{S}$ (return 1) or not (return 0). Then, we normalize $w_v$ regarding the maximum vertex importance weight,

$$w'_v(v_i) = \frac{w_v(v_i)}{\max w_v} \in [0, 1] \tag{5}$$

Finally, for a node $v_i$, its dropping probability is refined as follows,

$$p'_{dr}(v_i) = (1 - w'_v(v_i) \cdot \epsilon) \cdot p_{dr} \tag{6}$$

where $\epsilon \in (0, 1]$ specifies the maximum decay in the dropping probability for the node with the maximum importance weight. While Eq. 6 makes the dropping probability change linear to the node importance, we can set it to a general form,

$$p'_{dr}(v_i) = (1 - f(w'_v(v_i)) \cdot \epsilon) \cdot p_{dr} \tag{7}$$

where $f$ can be any monotonic increasing function with the input and output ranges defined on [0,1].

---

[2]This enhancement can work only for the probabilistic topology augmentation of node/edge-dropping. Many existing GCL methods have verified that node/edge dropping alone is enough for generating effective graph augmentations [24, 39, 52, 59, 60].

**Figure 1: Overview of the *CTAug* Framework. Module 1 enhances the probabilistic and deterministic augmentation process separately with the consideration of the cohesive subgraphs; Module 2 boosts GNN encoder to better capture the original graph's cohesion properties.**

For edge dropping augmentation, we calculate the dropping probability of an edge $e_{ij}$ by taking the average of the dropping probability of its two ends,

$$p'_{dr}(e_{ij}) = (p'_{dr}(v_i) + p'_{dr}(v_j))/2 \qquad (8)$$

#### 3.1.2 Deterministic Topology Augmentation.
Different from probabilistic augmentations, deterministic augmentation generates a *single* new graph from the original graph. As a representative, *MVGRL* [15] leverages a personalized PageRank [30] diffusion process to generate a deterministic augmented view from the original graph, which can be computed in a closed form [22]. In particular, the personalized PageRank diffusion can be calculated as,

$$S = \alpha(I - (1 - \alpha)D^{1/2}AD^{-1/2})^{-1} \qquad (9)$$

where $D$ is the diagonal degree matrix, $A$ is the adjacency matrix, and $\alpha$ denotes the teleport probability [22]. With *CTAug*, we can obtain a re-weighted adjacency matrix $A'$ where $A'_{i,j} = w'_e(e_{ij})$ (see Eq. 12). Then, we can use $A'$ to replace $A$ in Eq. 9 and conduct a cohesion-aware diffusion process.

As state-of-the-art deterministic topology augmentation strategies are mostly based on *graph diffusion*, e.g., Personalized PageRank and Markov Chain processes [15, 58], we then design an enhancement strategy to make the graph diffusion process cohesion-aware. The main idea is to assign larger weights to the graph edges in cohesive subgraphs so that the graph diffusion process would favor the large-weighted edges, as shown in Fig. 1.

We use $k$-core as an example to illustrate the process. First, given the original graph $\mathcal{G}$, we range $k$ from 1 to $k_{\max}$ and thus extract a set of $k$-core subgraphs $\mathbb{S} = \{\mathcal{S}_{core}^k | k = 1, 2, ..., k_{\max}\}$. Then, for a vertex $v_i$, we count how many times it appears in the set of subgraphs $\mathbb{S}$ to calculate its importance weight $w_v$.

$$w_v(v_i) = \sum_{\mathcal{S} \in \mathbb{S}} \mathbf{1}_{v_i \in vertex(\mathcal{S})} \qquad (10)$$

where $\mathbf{1}_{v_i \in vertex(\mathcal{S})}$ is an indicator function to output whether $v_i$ is in the vertex set of $\mathcal{S}$ (return 1) or not (return 0).

Then, we normalize $w_v$ regarding the average vertex importance weight,

$$w'_v(v_i) = \eta \cdot \frac{w_v(v_i)}{\bar{w}_v} + (1 - \eta) \cdot 1$$
$$\bar{w}_v = \frac{\sum_{v_i \in vertex(\mathcal{G})} w_v(v_i)}{|vertex(\mathcal{G})|} \qquad (11)$$

where $\eta \in [0, 1]$ is a factor controlling the degree to consider cohesive subgraphs. If $\eta$ is set to a value closer to 1, the cohesion property will be considered at a higher level.

Finally, suppose the original weight of edge $e_{ij}$ is $w_e(e_{ij})$, our updated weight $w'_e(e_{ij})$ is,

$$w'_e(e_{ij}) = \frac{1}{2}(w'_v(v_i) + w'_v(v_j))w_e(e_{ij}) \qquad (12)$$

Large vertex weights will increase the corresponding edge weights, and vice versa. We use the re-weighted graph as the input for deterministic augmentation (i.e., graph diffusion).

## 3.2 Graph Learning Enhancement

### 3.2.1 Subgraph-aware GNN Encoder.
While the topology augmentation enhancement part has ensured that the augmented view would probably retain cohesive subgraphs, the graph neural network (GNN) encoder may still lose this substructure information during the graph learning process. In general, conventional GNNs follow a message-passing neural network (MPNN) framework, as local information is aggregated and passed to neighbors [14, 29, 50]. Nevertheless, MPNNs have been proven to be limited in capturing subgraph properties, e.g., counting substructures [9]. Hence, we need to improve the GNN encoder's capacity to learn cohesive subgraph properties.

In *CTAug*, we propose an *original-graph-oriented graph substructure network (O-GSN)* to enhance existing GNN encoders, which is inspired by *graph substructure network (GSN)* [7]. GSN is a recently proposed topology-aware graph learning scheme to encode substructure information and is proven to be strictly more powerful than conventional GNNs. Specifically, GSN modifies the neighborhood aggregation process as,

$$\text{GSN: } AGG((\mathbf{h}_v, \mathbf{h}_u, \mathbf{s}_v, \mathbf{s}_u)_{u \in \mathcal{N}(v)}) \tag{13}$$

where $AGG$ is the neighborhood aggregation function such as $\sum_{u \in \mathcal{N}(v)} MLP(\cdot)$, $\mathbf{h}_v$ is the hidden state of node $v$, and $\mathbf{s}_v$ is the substructure-encoded feature of node $v$. In particular, $\mathbf{s}_v$ counts how many times node $v$ appears in a set of subgraph structures $\mathcal{H}$ (e.g., varying-size cliques). In brief, GSN adds an extra set of substructure-encoded node features to every GNN layer to enhance GNN's subgraph-aware ability. However, directly applying GSN into *CTAug* still faces two issues:

(i) *Low Efficiency*. GSN needs to learn $\mathbf{s}_v$ for every node in the graph with subgraph counting algorithms [11]. As the augmented view is randomly generated in GCL, directly applying GSN means that subgraph counting needs to be re-computed for every augmented view in an online manner, which is highly time-consuming.

(ii) *Losing Track of the Original Graph*. It is possible that two different original graphs generate the same augmented view. Directly applying GSN still cannot differentiate which original graph generates the augmented view.

To overcome the two issues, we propose the original-graph-oriented GSN, denoted as *O-GSN*. Specifically, *O-GSN* uses the substructure-encoded features from the original graph,

$$\text{O-GSN: } AGG((\mathbf{h}_v, \mathbf{h}_u, \mathbf{s}_v^o, \mathbf{s}_u^o)_{u \in \mathcal{N}(v)}) \tag{14}$$

where $\mathbf{s}_v^o$ is the substructure-encoded feature of node $v$ in the original graph.

With O-GSN, we only need to compute the substructure-encoded features for the original set of graphs in the data pre-processing stage, thus improving the training efficiency. Moreover, by considering features from the original graph, O-GSN enhances the encoder's power to differentiate the same augmented view from different original graphs, which may further enhance the GNN encoder's expressive power.

*Selection of Substructures in O-GSN*. In order to enhance the performance of GCL by considering cohesive subgraphs, the substructures selected in O-GSN should also be representative $k$-core/truss cohesive subgraphs. To achieve this, we analyze the cohesive properties of the candidate substructures used in the original GSN implementation and select those that are representative. In our current implementation, we focus on clique substructures. A detailed analysis of why we select cliques can be found in Appendix C.

### 3.2.2 Multi-Cohesion Embedding Fusion.
Since different cohesion properties can identify different important parts of graphs, one may want to take heterogeneous cohesion properties into account, e.g., both $k$-core and $k$-truss. To this end, we also design a *multi-cohesion embedding fusion* component to fuse embeddings obtained considering a set of various cohesion properties $\mathbb{C}$.

Specifically, we choose different cohesion properties and follow the augmentation enhancement process to train GNN encoders. Then, we concatenate embeddings learned from the augmentation strategy based on different cohesion properties as,

$$z_i = ||_{c \in \mathbb{C}} z_i^c \tag{15}$$

where $z_i \in \mathbb{R}^{n \times (d \cdot |\mathbb{C}|)}$ is the final graph embedding of $\mathcal{G}_i$, $z_i^c \in \mathbb{R}^{n \times d}$ is the graph embedding generated based on a certain cohesion property $c \in \mathbb{C}$, such as $k$-core/truss.

## 3.3 Extension for Node Embedding Learning

In general, the GCL methods for node embedding are often *local-local* GCL (comparison on node pairs) [58]. Representative local-local GCL methods include *GRACE* [59] and its follow-up *GCA* [60]. Their basic idea of augmentation is similar to *GraphCL*. First, two augmented views are generated with augmentation operations regarding certain probabilities. Afterward, the same nodes in two views are considered as a positive pair for node embedding learning.

As local-local GCL aims to learn node embedding, topology augmentation usually uses edge dropping in order to ensure that all the nodes still remain in the augmented view. Specifically, *GRACE* uses a randomized edge-dropping operation to generate augmented views; *GCA* improves *GRACE* by introducing a centrality-based adaptive edge dropping operation. Since this augmentation step is conceptually consistent with the edge dropping in *GraphCL*, we can use a similar procedure to enhance *GRACE* and *GCA*. It is worth noting that, since cohesion is a graph's substructure-level property, its importance to node embedding may not be as significant as to graph embedding.

## 4 HOW *CTAUG* POWERS GCL?

In this section, we provide a theoretical analysis of the performance of *CTAug* from the perspective of mutual information.[3] In particular, we analyze the topology augmentation enhancement module and the graph learning enhancement module separately. Detailed proofs for our analysis can be found in Appendix A.

We also conduct experiments to substantiate the efficacy mechanism of *CTAug*, and the detailed results are available in Appendix B.

---

[3]Note that we mainly focus on the contrastive schema between the original graph and the augmented graph, whereas the contrastive schema between two augmented graphs is analogous.

## 4.1 Topology Augmentation Enhancement

To begin, we introduce the definitions of sufficient encoder and minimal sufficient encoder, where $I$ represents mutual information.

**Definition 4.1.** [40] *(Sufficient Encoder) The encoder $f$ of $\mathcal{G}$ is sufficient in the contrastive learning framework if and only if $I(\mathcal{G}; \mathcal{G}') = I(f(\mathcal{G}); \mathcal{G}')$.*

The encoder $f$ is sufficient if the information in $\mathcal{G}$ about $\mathcal{G}'$ is lossless during the encoding procedure, which is required by the contrastive learning objective. Symmetrically, $I(\mathcal{G}; \mathcal{G}') = I(\mathcal{G}; f(\mathcal{G}'))$ if $f$ is sufficient.

**Definition 4.2.** [40] *(Minimal Sufficient Encoder) A sufficient encoder $f_1$ of $\mathcal{G}$ is minimal if and only if $I(f_1(\mathcal{G}); \mathcal{G}) \le I(f(\mathcal{G}); \mathcal{G})$, $\forall f$ that is sufficient.*

The minimal sufficient encoder only extracts relevant information about the contrastive task and discards irrelevant information.

**Theorem 4.3.** *Suppose $f$ is a minimal sufficient encoder. If $I(\mathcal{G}'; \mathcal{G}; y)$ increases, then $I(f(\mathcal{G}); y)$ will also increase.*

Given that cohesive properties are closely tied to the graph label $y$ [13, 16, 23], preserving more cohesive properties of the original graph $\mathcal{G}$ during graph augmentation (thereby increasing $I(y; \mathcal{G}; \mathcal{G}')$) enables the encoder $f$ to learn improved representations $f(\mathcal{G})$ through contrastive learning. This results in more retention of information related to $y$ for downstream tasks (i.e., enlarging $I(f(\mathcal{G}); y)$), so downstream task performance will elevate.

## 4.2 Graph Learning Enhancement

**Theorem 4.4.** *$f_1$ is our proposed O-GSN encoder with $k$-core ($k \ge 2$) or $k$-truss ($k \ge 3$) subgraphs considered in subgraph structures $\mathcal{H}$, $f_2$ is GIN (the default encoder). Then $I(f_1(\mathcal{G}); y) > I(f_2(\mathcal{G}); y)$.*

Based on *Theorem 4.4*, with other conditions kept constant, substituting the default GIN encoder with our proposed O-GSN encoder empowers the encoder to acquire enhanced representations through contrastive learning and preserve more information associated with $y$, which will boost the performance of downstream tasks.

## 5 EXPERIMENTS

### 5.1 Datasets and Settings

**Datasets**. We choose five social graph datasets [51] (*IMDB-B*, *IMDB-M*, *COLLAB*, *RDT-B*, *RDT-T*) and two biomedical graph datasets [6] (*ENZYMES*, *PROTEINS*). Table 1 summarizes the statistics.

- **IMDB-B** & **IMDB-M** [51] datasets contain actors/actresses' relations if they appear in the same movie. The label of each graph is the movie genre. In *IMDB-B*, the label is binary; in *IMDB-M*, the label is multi-class.
- **COLLAB** [51] is a scientific collaboration dataset. The researcher's ego network has three possible labels corresponding to the fields that the researcher belongs to.
- **RDT-B** [51] dataset includes user interaction graphs in *Reddit* threads, called *subreddits*. The task is to identify whether a subreddit graph is question/answer-based or discussion-based.
- **RDT-T** [35] dataset contains discussion and non-discussion based threads from *Reddit*. The task is to predict whether a thread is discussion-based or not.

- **ENZYMES** [6] includes proteins that are classified as enzymes or non-enzymes.
- **PROTEINS** [6] contains protein tertiary structures from 6 EC top-level classes.

**Experiment Setup**. We take the unsupervised representation learning setting commonly used for GCL benchmarks [58]. Following the evaluation scheme [43, 58], we train a linear SVM classifier based on graph embeddings for graph classification. We use 10-fold cross-validation and repeat each experiment five times. [4] Following most GCL studies in literature [54], we use accuracy to measure the graph classification performance.

**Hardware Environment**. Experiments are run on a server with a 28-core Intel CPU, 96GB RAM, and Tesla V100S GPU. The operating system is Ubuntu 18.04.5 LTS.

## 5.2 Methods

For graph classification tasks, we choose 7 GCL methods for graph-level representation learning as our baselines, including *GraphCL* [54], *JOAO* [53], *MVGRL* [15], *InfoGraph* [38], *AD-GCL* [39], *Auto-GCL* [52], and *RGCL* [24]. More details are in Appendix E.

To assess the effectiveness of *CTAug*, we apply it to enhance three GCL methods: two with probabilistic augmentations (*GraphCL* and *JOAO*) and one with deterministic augmentations (*MVGRL*). The resulting methods are denoted as **CTAug-GraphCL**, **CTAug-JOAO**, and **CTAug-MVGRL**, respectively. We consider two cohesion properties, namely $k$-core and $k$-truss, which we extracted from graphs using NetworkX[5] with the algorithms in [4, 10]. More details are in Appendix F.

## 5.3 Main Results

**Probabilistic GCL Method Enhancement (CTAug-GraphCL & CTAug-JOAO)**. Table 2 presents the graph classification results of several GCL methods. Among five social graph datasets, *IMDB-B*, *IMDB-M*, and *COLLAB* exhibit high average degrees ($\sim 10$ or larger). We expect that *CTAug* will perform well on these datasets, as high-degree graphs usually have highly-cohesive subgraphs.[6] Our experimental results validate this expectation. Specifically, *CTAug-GraphCL* yields an average accuracy improvement of 5.83% compared to *GraphCL* on three high-degree datasets. For *COLLAB*, the improvement is the most significant as *CTAug* can improve *GraphCL* by 9.36%, as this dataset has the largest average node degree ($\sim 65$). Similar to *GraphCL*, *CTAug* can also enhance *JOAO* by more than 5%.

For the remaining two social graph datasets, namely *RDT-B* and *RDT-T*, with low average degrees ($\sim 2$), *CTAug*'s performance improvement is marginal. The reason might be that *CTAug* primarily exploits the cohesion properties of a graph, and its effectiveness depends on the presence of highly cohesive substructures in the graph. *CTAug*'s improvements on biomedical graphs are also not as significant as the improvements on high-degree social graphs, since the average degrees of biomedical datasets are small ($\sim 3$).

---

[4] We fix random seeds to 1–5 for five cross-validation tests. Our results may look slightly different from the baselines' original papers due to different random seeds and evaluation schemes.
[5] https://networkx.org/
[6] Table 1 lists the average node degrees and the maximum value of $k$ in $k$-core/truss subgraphs ($k_{\max}$) for all the datasets.

**Table 1: Dataset statistics for graph classification.**

| Category | Dataset | #Graph | #Class | Avg. #Nodes | Avg. #Edges | Avg. Degree | Avg. $k_{max}$ ($k$-core) | Avg. $k_{max}$ ($k$-truss) |
|---|---|---|---|---|---|---|---|---|
| Social Graph | IMDB-B | 1,000 | 2 | 19.77 | 96.53 | 9.76 (high) | 9.16 | 10.16 |
| | IMDB-M | 1,500 | 3 | 13.00 | 65.94 | 10.14 (high) | 8.15 | 9.15 |
| | COLLAB | 5,000 | 3 | 74.49 | 2457.78 | 65.97 (high) | 40.53 | 41.52 |
| | RDT-B | 2,000 | 2 | 429.63 | 497.75 | 2.32 (low) | 2.33 | 3.09 |
| | RDT-T | 203,088 | 2 | 23.93 | 24.99 | 2.08 (low) | 1.58 | 2.46 |
| Biomedical Graph | ENZYMES | 600 | 6 | 32.63 | 62.14 | 3.81 (low) | 2.98 | 3.80 |
| | PROTEINS | 1,113 | 2 | 39.06 | 72.82 | 3.73 (low) | 3.00 | 3.83 |

**Table 2: Accuracy (%) on graph classification (OOM: out-of-memory).**

| Method | Social Graphs (High Degree) | | | | Social Graphs (Low Degree) | | | Biomedical Graphs | | |
|---|---|---|---|---|---|---|---|---|---|---|
| | IMDB-B | IMDB-M | COLLAB | AVG. | RDT-B | RDT-T | AVG. | ENZYMES | PROTEINS | AVG. |
| *InfoGraph* | 71.34±0.24 | 47.93±0.71 | 69.12±0.15 | 62.80 | 89.39±1.81 | 76.23±0.00 | 82.81 | 26.73±3.75 | 74.09±0.48 | 50.41 |
| *AD-GCL* | 71.28±1.10 | 47.59±0.62 | 71.22±0.89 | 63.36 | 88.84±0.90 | 76.51±0.00 | 82.68 | 27.33±2.28 | 73.39±0.85 | 50.36 |
| *AutoGCL* | 71.14±0.71 | 48.61±0.55 | 67.27±2.64 | 62.34 | 89.31±1.48 | 77.13±0.00 | 83.22 | 29.83±2.24 | 73.33±0.27 | 51.58 |
| *RGCL* | 71.14±0.64 | 48.28±0.60 | 73.48±0.93 | 64.30 | 91.38±0.40 | OOM | / | 33.33±1.61 | 73.37±0.35 | 53.35 |
| *GraphCL* | 71.48±0.44 | 48.11±0.60 | 72.36±1.76 | 63.98 | 91.69±0.70 | 77.44±0.03 | 84.57 | 32.83±2.05 | 74.32±0.76 | 53.58 |
| *CTAug-GraphCL* | 76.60±1.02 | 51.12±0.57 | 81.72±0.26 | 69.81 | **92.28±0.33** | **77.48±0.01** | **84.88** | 39.17±1.00 | 74.10±0.33 | 56.64 |
| *JOAO* | 71.40±0.38 | 48.68±0.36 | 73.40±0.46 | 64.49 | 91.66±0.59 | 77.24±0.00 | 84.45 | 34.60±1.06 | 74.32±0.46 | 54.46 |
| *CTAug-JOAO* | **76.80±0.71** | **51.19±0.88** | **81.90±0.53** | **69.96** | 92.19±0.24 | 77.35±0.02 | 84.77 | **39.92±1.36** | 74.46±0.13 | **57.19** |
| *MVGRL* | 71.88±0.73 | 50.19±0.40 | 80.48±0.29 | 67.52 | OOM | OOM | / | 34.20±0.67 | 74.33±0.62 | 54.27 |
| *CTAug-MVGRL* | 73.04±0.65 | 50.79±0.54 | 81.09±0.37 | 68.31 | OOM | OOM | / | 35.46±1.20 | **75.00±0.38** | 55.23 |

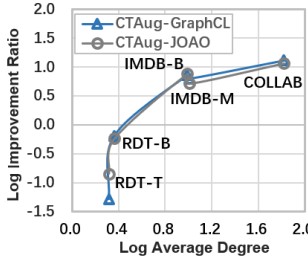

**Figure 2: *CTAug*'s improvement on datasets with varying average degrees.**

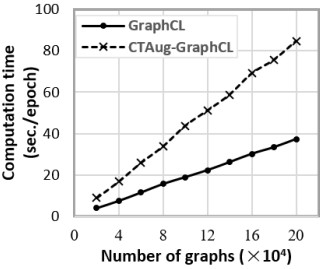

**Figure 3: Scalability test on RDT-T.**

**Table 3: Ablation study of *CTAug-GraphCL*.**

| Method | IMDB-B | IMDB-M | COLLAB | AVG. |
|---|---|---|---|---|
| *CTAug-GraphCL* | **76.60±1.02** | 51.12±0.57 | **81.72±0.26** | **69.81** |
| **Module Ablation** | | | | |
| *Only Module 1* | 71.54±0.27 | 49.11±0.48 | 72.64±0.63 | 64.43 |
| *Only Module 2* | 73.80±1.21 | 50.27±0.81 | 80.03±0.42 | 68.03 |
| **Cohesion Property Ablation** | | | | |
| *Only k-core* | 75.92±0.67 | **51.39±0.14** | 81.36±0.16 | 69.56 |
| *Only k-truss* | 76.12±1.20 | 50.99±0.57 | 80.71±0.30 | 69.27 |

Fig. 2 illustrates the performance enhancement achieved by *CTAug-GraphCL/JOAO* compared to *GraphCL/JOAO* across datasets with different average degrees. Notably, as the average degree increases, the impact of *CTAug* becomes more pronounced. We conclude that, before applying *CTAug*, it is prudent to ascertain whether the input graph is high-degree[7] or not.

**Deterministic GCL Method Enhancement (CTAug-MVGRL).** For deterministic GCL methods, *CTAug* can also boost the performance by comparing *CTAug-MVGRL* and *MVGRL*. Meanwhile, the improvement is minor even for high-degree graphs; the possible reason is that *MVGRL* has already used node degrees as features, which can be seen as a weak version of substructure-encoded features considered in Sec. 3.2.1 (as high-degree nodes are often in certain highly-cohesive subgraphs). Note that *MVGRL* cannot finish

training for large social graphs such as *RDT-B/T* due to out-of-memory. Hence, *CTAug-GraphCL/JOAO* may still be preferred for practical graph-level representation learning.

**Computation Scalability**. Fig. 3 shows how the computation time changes with the increase of training graphs. *CTAug-GraphCL* consumes about two times compared to *GraphCL* as *CTAug* trains graph representations considering both $k$-core and $k$-truss subgraphs; if only one subgraph property is considered, the training time overhead would be very small.

*CTAug* also needs to pre-compute cohesive subgraphs ($k$-core and $k$-truss in our implementation) for Module 1 and substructure-encoded features for Module 2 (O-GSN). The discovery of $k$-core and $k$-truss subgraphs for a single graph typically takes $\sim 10^{-2}$ seconds, while the computation of O-GSN features takes at most a few seconds (details are in Appendix G). Moreover, this procedure can be parallelized or conducted offline, allowing for the convenient integration of *CTAug* with a variety of existing methods.

---

[7]In accordance with our experimental results, graphs with average degree greater than 8 might be considered as high-degree graphs.

## 5.4 Ablation Study

We conduct experiments to evaluate the effectiveness of each module in *CTAug*, and the results are presented in Table 3. Since high-degree graphs are appropriate for *CTAug*, the ablation study is conducted on such graph datasets. As expected, using only one module of *CTAug* leads to a decrease in accuracy, which confirms the effectiveness of each module. While using only Module 2 has more improvements than using only Module 1, combining the two can enhance each other and achieve significantly higher accuracy. Previous studies have indicated that plain GNN cannot effectively learn subgraph properties [9], which may explain why using only Module 1 is not effective. Module 2 (O-GSN) assists GNN in preserving subgraph properties, thus enhancing Module 1.

We also examine the usefulness of combining multiple cohesion properties in our approach. However, we observe that fusion does not always improve accuracy. To gain more insight, we conducted an empirical analysis on the difference between $k$-core and $k$-truss subgraphs in *IMDB-B* and *IMDB-M*. Our findings show that the overlap between the $k$-core and $k$-truss subgraphs is larger than 95%, indicating that over 95% of nodes and edges are shared between the obtained subgraphs. This may explain why the performances of *CTAug* ($k$-core) and *CTAug* ($k$-truss) are close without much difference, and why fusion may sometimes even degrade performance. Future work may explore a more efficient fusion component to address this issue.

## 5.5 Parameter Analysis

Table 4 shows the performance of *CTAug-GraphCL/JOAO* when $\epsilon$ is varying. We observe that most settings of $\epsilon$ can increase accuracy compared to the original *GraphCL/JOAO*. The optimal choice for $\epsilon$ usually falls at 0.2, allowing for an appropriate trade-off between the diversity of augmented graphs (highest diversity at $\epsilon = 0$) and the preservation of cohesion properties (maximum preservation at $\epsilon = 1$).

**Table 4: Parameter analysis of $\epsilon$.**

| Method | $\epsilon$ | IMDB-B | IMDB-M | RDT-B |
|---|---|---|---|---|
| *GraphCL* | / | 71.48±0.44 | 48.11±0.60 | 91.69±0.70 |
| | 0.2 | 75.98±0.78 | 50.84±0.83 | 91.60±0.27 |
| | 0.4 | **76.60±1.02** | **51.12±0.57** | 91.88±0.32 |
| *CTAug-GraphCL* | 0.6 | 75.84±1.24 | 50.83±0.60 | 91.85±0.26 |
| | 0.8 | 75.68±0.70 | 50.16±0.24 | 91.94±0.41 |
| | 1.0 | 74.90±0.51 | 49.67±0.62 | **92.28±0.33** |
| *JOAO* | / | 71.40±0.38 | 48.68±0.36 | 91.66±0.59 |
| | 0.2 | **76.80±0.71** | **51.19±0.88** | **92.19±0.24** |
| | 0.4 | 76.36±1.42 | 50.48±0.83 | 92.17±0.30 |
| *CTAug-JOAO* | 0.6 | 76.56±0.49 | 50.40±0.88 | 91.52±0.54 |
| | 0.8 | 75.10±1.43 | 50.53±0.89 | 91.92±0.39 |
| | 1.0 | 75.18±1.29 | 50.17±0.75 | 92.01±0.42 |

## 5.6 Node Classification Results

We evaluate *CTAug* on two representative GCL methods for node embedding, namely *GRACE* [59] and *GCA* [60], referred to as *CTAug-GRACE* and *CTAug-GCA*, respectively. The node classification results of these methods on the *Coauthor-CS*, *Coauthor-Physics*, and

**Table 5: Results on node classification. The baseline results (except *GRACE* and *GCA*) are copied from [60] because we follow the same experimental setup. Meanwhile, we run *GRACE* and *GCA* by ourselves as we need to ensure that the exactly same configurations (neural network hidden units, training algorithm parameters, etc.) are used for *GRACE*/*GCA* and our enhanced *CTAug-GRACE*/*CTAug-GCA* for a fair comparison (OOM: out-of-memory).**

| Method | Coauthor CS | Coauthor Physics | Amazon Computers | AVG. |
|---|---|---|---|---|
| *DeepWalk+features* | 87.70±0.04 | 94.90±0.09 | 86.28±0.07 | 89.63 |
| *GAE* | 90.01±0.71 | 94.92±0.07 | 85.27±0.19 | 90.07 |
| *VGAE* | 92.11±0.09 | 94.52±0.00 | 86.37±0.21 | 91.00 |
| *DGI* | 92.15±0.63 | 94.51±0.52 | 83.95±0.47 | 90.20 |
| *GMI* | OOM | OOM | 82.21±0.31 | / |
| *MVGRL* | 92.11±0.12 | 95.33±0.03 | 87.52±0.11 | 91.65 |
| *GRACE* | 92.83±0.10 | 95.56±0.05 | 86.96±0.14 | 91.78 |
| *GCA* | 92.89±0.02 | 95.55±0.03 | 87.48±0.11 | 91.97 |
| *CTAug-GRACE* | 92.96±0.05 | **95.68±0.01** | 87.59±0.12 | 92.08 |
| *CTAug-GCA* | **92.98±0.04** | 95.61±0.01 | **88.30±0.13** | **92.30** |

*Amazon-Computers* datasets [37] are reported in Table 5. The dataset and baseline details are presented in Appendix H.

Our observations indicate that *CTAug-GRACE/GCA* yield some improvement over the original *GRACE/GCA*. However, the magnitude of this improvement is not as significant as the improvement of *CTAug* on graph classification tasks. This discrepancy may be attributed to the fact that cohesion is a subgraph property and therefore, more relevant to the entire graph than a single node.

Furthermore, as observed in graph classification, the improvement of *CTAug* is the most pronounced on *Amazon-Computers*, which has the highest degree (average degree is $\sim 35$ for *Amazon-Computers* and $\sim 10$ for the other two datasets). This reaffirms that *CTAug* is more effective for high-degree graphs, as these graphs generally contain more highly-cohesive substructures.

## 6 CONCLUSION AND LIMITATIONS

To introduce the awareness of cohesion properties (e.g., $k$-core and $k$-truss) into GCL, this work proposes a unified framework, called *CTAug*, that can be integrated with various existing GCL mechanisms. Two modules, including *topology augmentation enhancement* and *graph learning enhancement*, are designed to incorporate cohesion properties into the topology augmentation and graph learning processes of GCL, respectively. Extensive experiments have verified the effectiveness and flexibility of the *CTAug* framework.

Our current implementations are limited to $k$-core/truss cohesion properties, while many other types of properties are also crucial in practice. For instance, the average shortest path length is usually small for real-world social graphs such as *Facebook* [42]; then, it is reasonable to keep a small average shortest path length when augmenting social graphs. In the future, we will explore how to incorporate more graph properties into GCL procedures in a unified, flexible, and extensible manner.

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

# A PROOFS FOR THEORETICAL ANALYSIS (SEC. 4)

**Theorem 4.3.** *Suppose $f$ is a minimal sufficient encoder. If $I(\mathcal{G}'; \mathcal{G}; y)$ increases, then $I(f(\mathcal{G}); y)$ will also increase.*

PROOF. We denote $z = f(\mathcal{G})$, $z' = f(\mathcal{G}')$. $f$ is sufficient, so $I(\mathcal{G}; \mathcal{G}') = I(\mathcal{G}; z') = I(z; \mathcal{G}')$.

$$
\begin{aligned}
I(z; \mathcal{G}) &= H(z) & (z \text{ is a function of } \mathcal{G}) \\
&= I(z; \mathcal{G}') + H(z|\mathcal{G}') & (16) \\
&\geq I(z; \mathcal{G}') & (H(z|\mathcal{G}') \geq 0)
\end{aligned}
$$

Because $f$ is a minimal sufficient encoder, $I(z; \mathcal{G})$ will be minimized to $I(z; \mathcal{G}')$ and $H(z|\mathcal{G}') = 0$ holds.

$$
I(z; y) = I(z; z'; y) + I(z; y|z') \tag{17}
$$

$$
\begin{aligned}
I(z; z'; y) &= I(z; z'; y; \mathcal{G}) + I(z; z'; y|\mathcal{G}) \\
&= I(z; y; (z'; \mathcal{G})) + 0 & (z \text{ is a function of } \mathcal{G}) \\
&= I(z; y; (\mathcal{G}; \mathcal{G}')) & (I(\mathcal{G}; z') = I(\mathcal{G}; \mathcal{G}')) \\
&= I(y; \mathcal{G}; (z; \mathcal{G}')) \\
&= I(y; \mathcal{G}; (\mathcal{G}; \mathcal{G}')) & (I(\mathcal{G}'; z) = I(\mathcal{G}; \mathcal{G}')) \\
&= I(y; \mathcal{G}; \mathcal{G}')
\end{aligned}
\tag{18}
$$

$$
\begin{aligned}
I(z; y|z') &= I(z; y; \mathcal{G}'|z') + I(z; y|\mathcal{G}', z') \\
&= I(y; (z; \mathcal{G}')|z') + I(z; y|\mathcal{G}') & (z' \text{ is a function of } \mathcal{G}') \\
&= I(y; \mathcal{G}; z'|z') + I(z; y|\mathcal{G}') & (I(z; \mathcal{G}') = I(\mathcal{G}; z')) \\
&= 0 + I(z; y|\mathcal{G}') \\
&= 0 & (H(z|\mathcal{G}') = 0)
\end{aligned}
\tag{19}
$$

Based on Eq. 17, 18 and 19, $I(z; y) = I(y; \mathcal{G}; \mathcal{G}')$. As a result, the increase of $I(\mathcal{G}'; \mathcal{G}; y)$ leads to the growth of $I(f(\mathcal{G}); y)$.

From another perspective, we can extend InfoMin principle [40] to the graph field: the best-performing augmented graph should contain as much task-relevant information while discarding as much irrelevant information as possible. Formally, given the original graph $\mathcal{G}$ and its downstream task label $y$, the optimal augmented graph $\mathcal{G}'$ satisfies $I(\mathcal{G}; \mathcal{G}') = I(\mathcal{G}; y)$, which is called *sweet spot*.

If $I(y; \mathcal{G}; \mathcal{G}')$ increases, $I(\mathcal{G}; \mathcal{G}')$ will be close to $I(\mathcal{G}; y)$ (because their intersection is increasing), approaching sweet spot. So higher $I(y; \mathcal{G}; \mathcal{G}')$ indicates better-augmented graph $\mathcal{G}'$, i.e., $I(f(\mathcal{G}); y)$ will increase. We come to the same conclusion. □

**Lemma A.1.** *Given that $f$ is a GNN encoder with learnable parameters. Optimizing the loss function in Eq. 1 is equivalent to maximizing $I(f(\mathcal{G}); f(\mathcal{G}'))$, leading to the maximization of $I(f(\mathcal{G}); \mathcal{G}')$.*

PROOF. Appendix F in [54] provides theoretical justification that minimizing loss function Eq. 1 is equivalent to maximizing a lower bound of the mutual information between the latent representations of two augmented graphs, and can be viewed as one way of mutual information maximization between the latent representations. Consequently, the optimization of the loss function in Eq. 1 is equivalent to maximizing $I(f(\mathcal{G}); f(\mathcal{G}'))$.

Because $f(\mathcal{G})$ is a function of $\mathcal{G}$,

$$
\begin{aligned}
I(f(\mathcal{G}); \mathcal{G}') &= I(f(\mathcal{G}); f(\mathcal{G}'); \mathcal{G}') + I(f(\mathcal{G}); \mathcal{G}'|f(\mathcal{G}')) \\
&= I(f(\mathcal{G}); f(\mathcal{G}')) + I(f(\mathcal{G}); \mathcal{G}'|f(\mathcal{G}'))
\end{aligned}
\tag{20}
$$

Thus,

$$
I(f(\mathcal{G}); f(\mathcal{G}')) = I(f(\mathcal{G}); \mathcal{G}') - I(f(\mathcal{G}); \mathcal{G}'|f(\mathcal{G}')) \tag{21}
$$

While maximizing $I(f(\mathcal{G}); f(\mathcal{G}'))$, either $I(f(\mathcal{G}); \mathcal{G}')$ increases or $I(f(\mathcal{G}); \mathcal{G}'|f(\mathcal{G}'))$ decreases. When $I(f(\mathcal{G}); \mathcal{G}'|f(\mathcal{G}'))$ reaches it minimum value of 0, $I(f(\mathcal{G}); \mathcal{G}')$ will definitely increase. Hence, the process of maximizing $I(f(\mathcal{G}); f(\mathcal{G}'))$ can lead to the maximization of $I(f(\mathcal{G}); \mathcal{G}')$ as well. □

**Theorem 4.4.** *$f_1$ is our proposed O-GSN encoder with $k$-core ($k \geq 2$) or $k$-truss ($k \geq 3$) subgraphs considered in subgraph structures $\mathcal{H}$, $f_2$ is GIN (the default encoder). Then $I(f_1(\mathcal{G}); y) > I(f_2(\mathcal{G}); y)$.*

PROOF. Our proposed O-GSN is extended from GSN.

*Theorem 3.1* in [7] proves that if $H(\in \mathcal{H})$ is any graph except for star graphs, GSN is strictly more powerful[8] than MPNN. Apparently, $k$-core ($k \geq 2$) or $k$-truss ($k \geq 3$) graphs satisfy this condition. Thus, GSN is strictly more powerful than MPNN when $k$-core ($k \geq 2$) or $k$-truss ($k \geq 3$) subgraphs are considered in $\mathcal{H}$.

Despite different training processes, the graph embedding inference processes are the same for O-GSN and GSN, i.e., taking graph substructure features into consideration. Hence, O-GSN has the same ability as GSN to differentiate certain graphs that GIN (as an instance of MPNN) cannot differentiate [7]. That is, $f_1$ can capture more information of $\mathcal{G}$ than $f_2$,

$$
H(\mathcal{G}) \geq H(f_1(\mathcal{G})) > H(f_2(\mathcal{G})) \tag{22}
$$

$f_1(\mathcal{G})$ and $f_2(\mathcal{G})$ are functions of $\mathcal{G}$, so

$$
I(f_1(\mathcal{G}); \mathcal{G}) > I(f_2(\mathcal{G}); \mathcal{G}) \tag{23}
$$

$$
\begin{aligned}
I(f_1(\mathcal{G}); \mathcal{G}) &= I(f_1(\mathcal{G}); \mathcal{G}; \mathcal{G}') + I(f_1(\mathcal{G}); \mathcal{G}|\mathcal{G}') \\
&= I(f_1(\mathcal{G}); \mathcal{G}') - I(f_1(\mathcal{G}); \mathcal{G}'|\mathcal{G}) + I(f_1(\mathcal{G}); \mathcal{G}|\mathcal{G}') \\
&= I(f_1(\mathcal{G}); \mathcal{G}') + I(f_1(\mathcal{G}); \mathcal{G}|\mathcal{G}')
\end{aligned}
\tag{24}
$$

$$
I(f_1(\mathcal{G}); \mathcal{G}') = I(f_1(\mathcal{G}); \mathcal{G}) - I(f_1(\mathcal{G}); \mathcal{G}|\mathcal{G}') \tag{25}
$$

In Eq. 24, because $f_1(\mathcal{G})$ is a function of $\mathcal{G}$, $I(f_1(\mathcal{G}); \mathcal{G}'|\mathcal{G}) = 0$. According to *Lemma A.1*, during the contrastive learning process, our optimization objective is to maximize $I(f_1(\mathcal{G}); \mathcal{G}')$, so $I(f_1(\mathcal{G}); \mathcal{G}|\mathcal{G}')$ is approaching its minimum value of 0. Hence,

$$
I(f_1(\mathcal{G}); \mathcal{G}) \approx I(f_1(\mathcal{G}); \mathcal{G}') \tag{26}
$$

Similarly,

$$
I(f_2(\mathcal{G}); \mathcal{G}) \approx I(f_2(\mathcal{G}); \mathcal{G}') \tag{27}
$$

Combining Eq. 23, 26 and 27, we get

$$
I(f_1(\mathcal{G}); \mathcal{G}') > I(f_2(\mathcal{G}); \mathcal{G}') \tag{28}
$$

---

[8] *expressive power* means the ability of the GNN model to capture and represent complex patterns and information within a graph structure [50].

$$I(f_1(\mathcal{G}); \mathcal{G}') = I(f_1(\mathcal{G}); \mathcal{G}'; y) + I(f_1(\mathcal{G}); \mathcal{G}'|y)$$
$$= I(f_1(\mathcal{G}); y) - I(f_1(\mathcal{G}); y|\mathcal{G}') + I(f_1(\mathcal{G}); \mathcal{G}'|y) \tag{29}$$

$$I(\mathcal{G}'; \mathcal{G}|y) = I(f_1(\mathcal{G}); \mathcal{G}'; \mathcal{G}|y) + I(\mathcal{G}'; \mathcal{G}|y, f_1(\mathcal{G}))$$
$$\geq I(f_1(\mathcal{G}); \mathcal{G}'; \mathcal{G}|y) \quad \text{(the non-negativity of } I) \tag{30}$$
$$= I(f_1(\mathcal{G}); \mathcal{G}'|y) \quad (f_1(\mathcal{G}) \text{ is a function of } \mathcal{G})$$

According to *Lemma A.1*, our optimization objective is to maximize $I(f_1(\mathcal{G}); \mathcal{G}')$ in the contrastive learning process. Therefore, $I(f_1(\mathcal{G}); y|\mathcal{G}')$ approaches its minimum value of 0 and $I(f_1(\mathcal{G}); \mathcal{G}'|y)$ is nearing its maximum value of $I(\mathcal{G}'; \mathcal{G}|y)$.

$$I(f_1(\mathcal{G}); \mathcal{G}') \approx I(f_1(\mathcal{G}); y) + I(\mathcal{G}'; \mathcal{G}|y) \tag{31}$$

Similarly,

$$I(f_2(\mathcal{G}); \mathcal{G}') \approx I(f_2(\mathcal{G}); y) + I(\mathcal{G}'; \mathcal{G}|y) \tag{32}$$

Combining Eq. 28, 31 and 32, we get

$$I(f_1(\mathcal{G}); y) > I(f_2(\mathcal{G}); y) \tag{33}$$

$\square$

## B  EMPIRICAL ANALYSIS DETAILS (SEC. 4)

To validate the efficacy mechanism of *CTAug* empirically, we initially confirm the crucial significance of graph cohesion properties for downstream tasks (e.g., graph classification). Subsequently, we verify that *CTAug*'s topology augmentation enhancement module can preserve the cohesion properties of the original graph to a greater extent during graph augmentation. Finally, we validate that the graph learning enhancement module ensures that the GNN encoder also acquires the cohesion property information and incorporates it into graph embedding.

**Effectiveness of Cohesion Properties.** To verify the connection between cohesion properties and graph labels, we convert cohesive subgraphs into graph features and train the same SVM classifier as our graph classification evaluation experiments. To be specific, the $i$-th $k$-core feature of a graph $\mathcal{G}$ is the number of nodes in its $i$-core subgraph, and the $i$-th $k$-truss feature is the number of nodes in its $i$-truss subgraph.

Table 6 presents the classification results of the above feature construction method, considering cohesive subgraphs. It is evident that the inclusion of cohesive features leads to a substantial enhancement in classification accuracy compared to random selection, particularly in high-degree graphs like *COLLAB*, where accuracy more than doubles. Consequently, we can deduce that cohesive properties exhibit a strong correlation with graph labels, so incorporating these properties into our graph contrastive learning process provides valuable priors.

**Effectiveness of Module 1 (Topology Augmentation Enhancement).** Table 7 demonstrates that the node drop augmentation of our *CTAug* method effectively preserves more nodes in cohesive subgraphs and retains cohesion property in the augmented graphs, compared with random node drop augmentation (used in *GraphCL*). This aligns with the design goals of *CTAug*'s Module 1.

**Effectiveness of Module 2 (Graph Learning Enhancement).** The ablation study in Table 3 (Sec. 5.4) shows that the removal of Module 2 leads to a significant decrease in classification accuracy. This observation validates that Module 2 effectively empowers the GNN encoder to incorporate more cohesion information into the graph embedding.

**Conclusion.** Our experimental findings confirm: (1) there is a strong correlation between cohesion properties and downstream tasks; (2) Module 1 of *CTAug* succeeds in producing cohesion-preserving augmented graphs; (3) Module 2 enhances the capture of cohesion properties during representation learning. Therefore, *CTAug* effectively captures cohesion information of the original graph and is poised to improve performance in downstream tasks.

## C  SUBSTRUCTURE SELECTION DETAILS FOR O-GSN (SEC. 3.2.1)

We use the classic graphs generators of NetworkX to get a set of substructures, such as cycle, clique, and path graphs, which are also considered in the original GSN implementation [7]. Specifically, we select cliques for our implementation in O-GSN, as they constitute the majority of the $k_{\max}$-core/truss subgraphs in the datasets we examined. For instance, in IMDB-B and IMDB-M, we observed that over 80% of $k_{\max}$-core/truss subgraphs are cliques (where $k_{\max}$ represents the maximum $k$-core/truss subgraph). Similarly, in RDT-B, the percentage of $k_{\max}$-core/truss subgraphs containing a clique is larger than 75%. Our empirical analysis further confirms that cliques outperform other substructures, and selecting 3/4/5 together generally leads to better results compared to selecting only one of them (Table 9).

## D  $k_{\max}$ DISTRIBUTION FOR DATASETS (SEC. 5.1)

Fig. 4 depicts the distribution of $k_{\max}$, the maximum $k$-core index, for different datasets. We can observe that the *IMDB-B/M* and *COLLAB* datasets have higher degrees and $k_{\max}$ values, indicating the presence of more highly cohesive subgraphs. Therefore, we anticipate that *CTAug* may achieve better performance on these three datasets.

## E  BASELINE METHODS DETAILS (SEC. 5.2)

- **InfoGraph** [38] maximizes the mutual information between graph-level representations and different scales' sub-structure-level representations to learn graph embedding without graph augmentations. We run *InfoGraph* with the *PyGCL* library.[9]
- **MVGRL** [15] uses personalized PageRank on the original graph to generate a diffusion matrix as the augmented view for GCL.[10]
- **GraphCL** [54] designs four types of graph augmentations (random node dropping/edge perturbation/attribute masking/random walk-based subgraph sampling) used for GCL. We run *GraphCL* with the *PyGCL* library.

---

[9]https://github.com/PyGCL/PyGCL
[10]https://github.com/kavehhassani/mvgrl

**Table 6: Accuracy (%) on graph classification with linear SVM classifier.**

| Input Feature | Considering Cohesion | Social Graphs (High Degree) | | | | Social Graphs (Low Degree) | | | Biomedical Graphs | | |
|---|---|---|---|---|---|---|---|---|---|---|---|
| | | IMDB-B | IMDB-M | COLLAB | AVG. | RDT-B | RDT-T | AVG. | ENZYMES | PROTEINS | AVG. |
| *None (random selection)* | ✗ | 50.60±5.62 | 33.27±2.79 | 32.48±3.06 | 38.78 | 50.20±3.08 | 50.03±0.39 | 50.12 | 14.17±2.81 | 48.43±5.08 | 31.30 |
| *k-core node count* | ✔ | 69.90±3.53 | 49.73±3.44 | 76.28±2.11 | 65.30 | 79.90±2.75 | 63.50±0.30 | 71.70 | 25.67±6.06 | 74.38±3.80 | 50.03 |
| *k-truss node count* | ✔ | 69.80±3.82 | 49.47±3.66 | 76.04±2.13 | 65.10 | 78.25±3.34 | 63.06±0.32 | 70.66 | 28.83±6.58 | **74.83±2.99** | 51.83 |
| *k-core & k-truss node count* | ✔ | 69.60±3.69 | 49.53±3.58 | 76.92±1.86 | 65.35 | 80.80±2.83 | 64.15±0.32 | 72.48 | 30.33±5.26 | 74.38±4.12 | 52.36 |
| *GraphCL embedding* | ✗ | 71.48±0.44 | 48.11±0.60 | 72.36±1.76 | 63.98 | 91.69±0.70 | 77.44±0.03 | 84.57 | 32.83±2.05 | 74.32±0.76 | 53.58 |
| *CTAug-GraphCL embedding* | ✔ | **76.60±1.02** | **51.12±0.57** | **81.72±0.26** | **69.81** | **92.28±0.33** | **77.48±0.01** | **84.88** | **39.17±1.00** | 74.10±0.33 | **56.64** |

**Table 7: Proportion of cohesive subgraph nodes preserved in the augmented graph on average. We set node dropping probability $p_{dr} = 0.2$ and decay factor $\epsilon = 0.2$.**

| Augmentation | Property | IMDB-B | IMDB-M | COLLAB | RDT-B | RDT-T | ENZYMES | PROTEINS | AVG. |
|---|---|---|---|---|---|---|---|---|---|
| *Random node drop* | $k$-core | 0.801 | 0.799 | 0.802 | 0.800 | 0.800 | 0.800 | 0.799 | 0.800 |
| | $k$-truss | 0.803 | 0.801 | 0.800 | 0.800 | 0.800 | 0.801 | 0.798 | 0.800 |
| *CTAug node drop* | $k$-core | 0.837 | 0.838 | 0.840 | 0.825 | 0.833 | 0.837 | 0.835 | 0.835 |
| | $k$-truss | 0.836 | 0.838 | 0.839 | 0.825 | 0.833 | 0.832 | 0.827 | 0.833 |

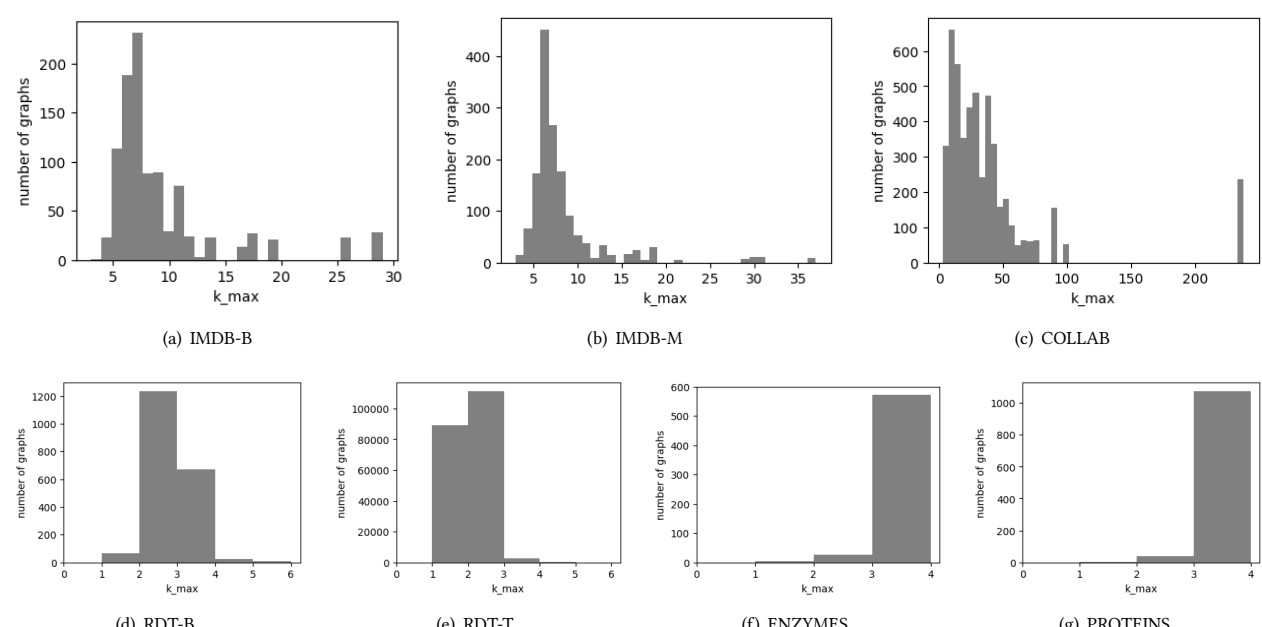

(a) IMDB-B        (b) IMDB-M        (c) COLLAB

(d) RDT-B   (e) RDT-T   (f) ENZYMES   (g) PROTEINS

**Figure 4: Histogram of $k_{\max}$ ($k$-core).**

- **JOAO** [53] extends *GraphCL* by adaptively choosing the augmentation operation. We re-implement *JOAO* based on *PyGCL* for experimentation.[11]
- **AD-GCL** [39] optimizes graph augmentations in an adversarial way to give encoder the minimal sufficient information.[12]
- **AutoGCL** [52] builds learnable generative node-wise augmentation policies for graph contrastive learning in an end-to-end manner.[13]
- **RGCL** [24] automatically discovers rationales as graph augmentations.[14]

---

[11]We also tried the code released by [53] in https://github.com/Shen-Lab/GraphCL_Automated, but the results are worse than our re-implementation. So we report the results with our re-implementation.
[12]https://github.com/susheels/adgcl

[13]https://github.com/Somedaywilldo/AutoGCL
[14]https://github.com/lsh0520/RGCL

**Table 8: Relationship between classic substructures and $k$-core/$k$-truss.**

| Substructure | $k$-core | $k$-truss |
|---|---|---|
| $k$-cycle | 2-core | / |
| $k$-clique | $(k-1)$-core | $k$-truss |
| $k$-path | 1-core | / |
| $k$-star | 1-core | / |
| $k$-binomial-tree | 1-core | / |
| $k$-nonisomorphic-trees | 1-core | / |

**Table 9: Substructure and $k$ selection (*CTAug-GraphCL*).**

| Substructure | k | IMDB-B | IMDB-M | AVG. |
|---|---|---|---|---|
| clique | 3 | 75.82±0.43 | 50.65±0.52 | 63.24 |
| clique | 4 | 75.65±0.34 | **51.22±0.41** | 63.44 |
| clique | 5 | 75.10±0.37 | 50.53±0.22 | 62.82 |
| clique | 3,4,5 | **76.60±1.02** | 51.12±0.57 | **63.86** |
| clique | 4 | **75.65±0.34** | **51.22±0.41** | **63.44** |
| cycle | 4 | 74.18±0.47 | 49.17±0.45 | 61.68 |
| star | 3 | 70.45±1.13 | 48.98±0.68 | 59.72 |
| path | 4 | 67.25±0.43 | 47.80±0.64 | 57.53 |
| binomial-tree | 2 | 67.25±0.43 | 47.80±0.64 | 57.53 |

**Table 10: Probabilistic topology augmentation behaviors of existing GCL methods.**

| Method | Aug. Operation | Aug. Probability |
|---|---|---|
| GraphCL [54] | randomly selected | uniform |
| JOAO [53] | min-max optimized | uniform |
| AD-GCL [39] | edge dropping | adversarial learning |
| AutoGCL [52] | node dropping | generative learning |
| RGCL [24] | node dropping | rationale-based learning |

Table 10 shows the comparison of augmentation operations and probabilities of existing probabilistic GCL topology augmentation methods.

## F  IMPLEMENTATION DETAILS FOR *CTAUG* VARIANTS (SEC. 5.2)

Here, we clarify the implementation details of our methods in experiments. We implement our method with Python 3.8 and PyTorch 1.12.0.

(1) **CTAug-GraphCL.** *GraphCL* [54] randomly selects an operation from $\mathcal{T}$ ={*node dropping, edge dropping, edge adding, random walk-based sampling*} and then performs the probabilistic augmentation in a uniform manner (e.g., every node has the same probability of being removed). We fix the augmentation operation to node dropping (with the default probability of 0.2), as node dropping proves to be generally well across different datasets [24, 54]. The default GNN encoder of *GraphCL*, i.e., GIN [50], is then enhanced by O-GSN to consider cohesive-substructure features. We implement this method mainly based on PyGCL. The hidden dim is 128, and the batch size is chosen from {16, 64} according to the size of the graphs.

Table 11 presents the graph classification performance for different $f$ functions in Eq. 7. Overall, there are no substantial differences between different functions. We select $f(x) = x^2$ in our implementation. We set the dropping probability decay factor $\epsilon$ through grid search for each dataset. Table 12 shows the grid search results.

**Table 11: Accuracy(%) on graph classification for different $f$ functions.**

| | IMDB-B | IMDB-M | AVG. |
|---|---|---|---|
| GraphCL | 71.48±0.44 | 48.11±0.60 | 59.80 |
| + CTAug ($f(x) = x$) | **76.85±1.60** | 50.98±0.62 | **63.92** |
| + CTAug ($f(x) = \sqrt{x}$) | 76.12±1.10 | **51.42±0.75** | 63.77 |
| + CTAug ($f(x) = x^2$) | 76.60±1.02 | 51.12±0.57 | 63.86 |

(2) **CTAug-JOAO.** In *CTAug-JOAO*, both node and edge-dropping operations are kept. The usage of the node or edge-dropping is determined by the optimization algorithm in *JOAO* [53]. Other parameter settings are the same as **CTAug-GraphCL**.

(3) **CTAug-MVGRL.** We implement this framework mainly based on MVGRL. We use GCN as the encoder, and the number of hidden units is 128. The batch size is 64. The factor $\eta$ controlling the degree to consider cohesive subgraphs in Eq. 11 is also set with grid search on the specific dataset, and the grid search results are shown on Table 12.

(4) **CTAug-GRACE & GTAug-GCA.** We implement this framework based on GRACE and GCA. We select degree centrality as the centrality measure, and the parameter settings are the same as in the original GRACE [59] and GCA [60] papers. The dropping probability decay factor $\epsilon$ is fixed at 1. The function $f$ in Eq. 7 is instantiated as $f(x) = x$.

It should be noted that for computational resource and performance reasons, we only employ Module 1 for the *CTAug-GraphCL* and *CTAug-JOAO* methods on the *RDT-B* dataset, as well as for the *CTAug-MVGRL* method on the *COLLAB*, *ENZYMES*, and *PROTEINS* datasets.

## G  PRE-COMPUTATION TIME (SEC. 5.3)

Table 13 presents the average pre-computation time for one graph. For the computation of O-GSN features for high-degree graphs, it takes up to a few seconds, while low-degree graphs only require $0.00X$ seconds. Calculating $k$-core and $k$-truss subgraphs typically only takes $0.00X \sim 0.0X$ seconds. We take 3-clique as substructure to calculate O-GSN features for *COLLAB*, and we choose 3,4,5-clique for other datasets.

## H  DATASETS AND BASELINES FOR NODE CLASSIFICATION (SEC. 5.6)

For *node classification*, we conduct experiments on *Coauthor-CS*, *Coauthor-Physics* and *Amazon-Computers* (Table 14).

- **Coauthor-CS** & **Coauthor-Physics** [37] are two co-authorship graphs based on the Microsoft Academic Graph from the KDD Cup 2016 challenge. In these graphs, nodes are authors; node features represent paper keywords for each author's papers; edges reveal co-authorship relationships; class labels indicate their most active research field.

### Table 12: Specific factor values obtained by grid search.

| Parameter | IMDB-B | IMDB-M | COLLAB | RDT-B | RDT-T | ENZYMES | PROTEINS |
|---|---|---|---|---|---|---|---|
| $\epsilon$ (CTAug-GraphCL) | 0.2 | 0.4 | 0.2 | 0.4 | 0.2 | 0.4 | 0.8 |
| $\epsilon$ (CTAug-JOAO) | 0.2 | 0.2 | 0.2 | 0.2 | 0.2 | 0.2 | 1.0 |
| $\eta$ (CTAug-MVGRL) | 0.4 | 0.4 | 0.2 | / | / | 0.6 | 0.8 |

### Table 13: Average pre-computation time (seconds per graph).

| Precomputation | IMDB-B | IMDB-M | COLLAB | RDT-B | RDT-T | ENZYMES | PROTEINS |
|---|---|---|---|---|---|---|---|
| O-GSN features | 5.357 | 3.868 | 4.355 | 0.005 | 0.001 | 0.002 | 0.003 |
| $k$-core subgraphs | 0.001 | 0.001 | 0.015 | 0.007 | 0.000 | 0.001 | 0.001 |
| $k$-truss subgraphs | 0.001 | 0.001 | 0.081 | 0.009 | 0.000 | 0.001 | 0.001 |

### Table 14: Dataset statistics for node classification task.

| Dataset | #Nodes | #Edges | #Features | #Classes | Avg. Degree | $k_{\max}$ ($k$-core) | $k_{\max}$ ($k$-truss) |
|---|---|---|---|---|---|---|---|
| *Coauthor-CS* | 18,333 | 81,894 | 6,805 | 15 | 8.93 | 19 | 20 |
| *Coauthor-Physics* | 34,493 | 247,962 | 8,415 | 5 | 14.38 | 18 | 12 |
| *Amazon-Computers* | 13,752 | 245,861 | 767 | 10 | 35.76 | 53 | 33 |

- **Amazon-Computers** [37] is a co-purchase relationship network built based on Amazon, where nodes represent goods, and two goods are connected if customers frequently buy them together. Each node has a bag-of-words feature (encoding product reviews), and class labels indicate the product category.

We use eight representative baseline methods that learn node embedding in an unsupervised manner. Node features are considered in all the baselines.

- **DeepWalk** [33] uses local information obtained from random walks to learn latent representations without supervision. Note that the original *DeepWalk* does not consider node features. To make a fair comparison, we concatenate a node's *DeepWalk*-learned embedding and raw features together as a node's representation.

- **GAE, VGAE** [19] uses latent variables to learn representations for graphs based on variational auto-encoders.
- **DGI** [44] maximizes the mutual information between patch representations and high-level summaries of graphs.
- **GMI** [32] maintains the consistency of information between the input and output of a graph neural encoder.
- **MVGRL** [15] contrasts encodings from first-order neighbors and a graph diffusion.
- **GRACE** [59] generates two graph views by corruption and learn node representations by maximizing the similarity between these two views' node representations.
- **GCA** [60] augments the original graph adaptively by incorporating centrality priors (e.g., degree centrality).

Received 20 February 2007; revised 12 March 2009; accepted 5 June 2009

