# OpenReview forum: "Graph Contrastive Learning with Cohesive Subgraph Awareness"
_ACM.org/TheWebConf/2024/Conference — TheWebConf24_

### Official Review · Reviewer_vsR3 · 2023-11-21

**Novelty:** 5
**Technical Quality:** 6

**Review:**

Contributions:

Motivated by the desire to protect cohesive graph properties against graph augmentation during graph contrastive learning (GCL), the authors propose a framework called CTAug that is cohesion-aware. CTAug retains the most
cohesive subgraphs from the original graph (e.g., k-clique, k-core) when generating augmented views. CTAug comprises topology augmentation (to preserve cohesion) and graph learning (to discern subgraphs) enhancement modules. The authors aim to show that CTAug theoretically and empirically improve SOTA performance on graph and node representation learning.

My recommendation is based on S1, S2, S3, W1, W2, W3. I am happy to raise my scores based on the authors' responses to my questions and clarification/justification of W1, W2, W3.

Quality:

Pros:
- The authors provide thorough justification of why cohesive substructures are relevant to downstream tasks (Section 2.1).

- CTAUG is simple and intuitive.

- (S1) Experiments: The authors run experiments on a variety of datasets and relevant baselines. The usage of CTAUG significantly increases graph classification accuracy on all the datasets. The ablation experiment results (that both augmentation and Graph Substructure Network are needed) are convincing. CTAUG achieves comparable accuracy to baselines on node classification datasets.

Cons:
- The authors should analyze the probability and extent to which cohesive subgraphs are actually preserved with probabilistic topology augmentation. They can do this theoretically, and empirically using the datasets with which they experiment.

- The authors should comment (if possible) on how cohesive subgraphs are relevant in the context of each graph learning task/dataset they consider.

- Line 594: If $f_1$ is sufficient, then is it ever possible $I(f_1(G), G) = I(G, G) < I(f(G), G) = I(G, G)$ for another sufficient encoder $f$? Per definition 4.1, it appears that all sufficient encoders are minimally sufficient.

- (W3) Theorem 4.3: If $f$ is not a minimal sufficient encoder, will $I(f(G), y)$ still increase?

- Theorem 4.4: While GSN is more expressive than GIN, its ability to distinguish relevant substructures is contingent on learning appropriate parameters [1].

Clarity:

Pros:
- (S3) The writing is generally clear and well-organized.

Originality:

Pros:
- (S2) CTAUG can be flexibly applied to existing stochastic and deterministic GCL mechanisms to consider cohesive subgraphs.

Cons:
- (W1) CTAUG is a relatively simple combination of link re-weighting heuristics and Graph Substructure Network.

Significance:

Pros:
- The authors address the important issue that augmented views of graphs may not preserve important components of the graph (e.g., k-core).

Cons:
- (W2) The authors should elaborate further on the limitations of their approach, e.g., in which situations may cohesive subgraphs *not* be relevant (beyond graphs with low-degree nodes)?

Miscellaneous suggestions:
- The authors could investigate how CTAUG impacts graph/node classification precision/recall.

[1] Xu, Keyulu, et al. "How powerful are graph neural networks?." arXiv preprint arXiv:1810.00826 (2018).

EDIT: I have read the authors' rebuttal.

**Questions:**

Please see Review (above).

**Reviewer Confidence:**

3: The reviewer is confident but not certain that the evaluation is correct

**Scope:**

3: The work is somewhat relevant to the Web and to the track, and is of narrow interest to a sub-community

---

### Official Review · Reviewer_mfRv · 2023-11-24

**Novelty:** 3
**Technical Quality:** 4

**Review:**

Summary:
This paper presents a new graph contrastive learning model with cohesive subgraph awareness. Specifically, the proposed CTAug improves over the existing works in the following ways: (1) it presents a topology augmentation module tailored for cohesive subgraphs, (2) it leverages both deterministic and probabilistic augmentations, and (3) it presents an enhanced graph encoders to consider subgraph features.

Strengths
+ The idea is clearly presented and is easy to understand.
+ The overall framework appears to be technically sound.

Weaknesses:
- The technical contribution is limited. Apart from the cohesive subgraph extraction, the other two modules are not clearly motivated. For example, it is not clear to me what are "Unified Framework" and "Expressive Network" in the introduction. Also, the whole model heavily relies on the assumption that "cohesive properties are closely tied to the graph label y", which might not be the case for all kinds of graphs (say graphs with high heterophily). In this case, the authors need to clearly demonstrate the scope where their model is applicable.
- The theoretical justification appears to be weak as well. The proof mostly follows from [40] which has nothing related to do with graphs. Also, the authors **do** need to show the proof for the contrastive schema between two augmented graphs, as this is what they claimed in the "Unified Framework" part, which involves deterministic and probabilistic augmentations separately.
- Experimental evaluation is also problematic with no recent baselines included (for example, Sub-graph Contrast for Scalable Self-Supervised Graph Representation Learning). The authors also need to clarify whether their framework is stronger than graph contrastive learning models without explicit augmentations (for example, SimGRACE: A Simple Framework for Graph Contrastive Learning without Data Augmentation).

**Questions:**

Please see the weaknesses part in the above section.

=========================================
Post rebuttal feedback:
I believe most of concerns have been cleared and I am willing to increase my score.

**Reviewer Confidence:**

4: The reviewer is certain that the evaluation is correct and very familiar with the relevant literature

**Scope:**

3: The work is somewhat relevant to the Web and to the track, and is of narrow interest to a sub-community

---

### Official Review · Reviewer_gFhZ · 2023-11-24

**Novelty:** 4
**Technical Quality:** 4

**Review:**

This paper proposes a novel unified framework called CTAug, to seamlessly integrate cohesion awareness into various existing GCL mechanisms. In particular, CTAug comprises two specialized modules: topology augmentation enhancement and graph learning enhancement. The former module generates augmented graphs that carefully preserve cohesion properties, while the latter module bolsters the graph encoder’s ability to discern subgraph patterns.
Strengths:
Theoretical analysis from the perspective of mutual information shows that CTAug can improve the existing GCL mechanism, especially for graphs with high degrees
Weaknesses:
--The author appears to have merely transposed an existing method into the realm of comparative learning, lacking a pronounced sense of novelty. The employed structure closely resembles MVGRL. However, a point of departure is identified in the utilization of 𝑘-core and 𝑘-truss within the data augmentation segment to attain subgraph data enhancement.
--The effectiveness of data augmentation in various categorical datasets, specifically regarding the persistence of 𝑘-truss and 𝑘-core on datasets of different types, has not been elucidated in the paper. This lack of clarification is noteworthy, given that the title of the paper suggests a universal approach to data augmentation.
--For diverse datasets, the modules of the Cohesive Subgraph Awareness model require sufficient flexibility during the learning process, and they have not adequately taken into account other types of attributes.
--The author did not provide a comparison with some of the most recent methods for comparative learning data augmentation.

**Questions:**

--While the primary focus of data augmentation is on topology, one might contemplate integrating structural data augmentation to enhance overall effectiveness. (It is noteworthy that the experimental outcomes for the novel unsupervised method do not appear highly promising, as indicated by the results presented in https://paperswithcode.com/sota/graph-classification-on-collab.)

-- As delineated in the ablation section, the abatement of Cohesion Property raises questions about the current efficiency of the model design. Does the utilization of solely the second module in Module Ablation seem to exert a diminished influence on accuracy?
--Random data augmentation may indeed undermine the semantic information of the graph. However, the data augmentation methods 𝑘-core and 𝑘-truss, as discussed in the paper, have attracted attention well before their application in social networks. I am uncertain about the universal applicability of these methods as effective data augmentation techniques across all types of graphs, especially in the case of graphs containing noise. How does the performance of these methods manifest in graphs with noise?

**Reviewer Confidence:**

3: The reviewer is confident but not certain that the evaluation is correct

**Scope:**

3: The work is somewhat relevant to the Web and to the track, and is of narrow interest to a sub-community

---

### Official Review · Reviewer_nEpW · 2023-11-30

**Novelty:** 5
**Technical Quality:** 5

**Review:**

This paper utilizes cohesive topological properties to generate graph augmentation and proposes a Subgraph-aware GNN Encoder to encode subgraphs, which is a novel approach.
The paper provides a detailed description of the model architecture and follows a logical structure which is easy to understand.

The experiments in this paper are comprehensive. Both the classification experiments and ablation studies demonstrate the effectiveness of the proposed method.

**Questions:**

1.	The purpose of using Cohesive Subgraph to generate augmentation is not clear, utilizing cohesive topological properties seems to be not a general approach for all graphs, rather a method for dealing with specific graphs.
2.	Equations 13 and 14 lack information; it's not evident how to perform the aggregation. I hope these two equations can offer a clearer description.
3.	Lack of some SOTA baselines for both node and graph classification[ 1]
Spectral Augmentation for Self-Supervised Learning on Graphs

**Reviewer Confidence:**

3: The reviewer is confident but not certain that the evaluation is correct

**Scope:**

4: The work is relevant to the Web and to the track, and is of broad interest to the community

---

### Decision · Program_Chairs · 2024-01-22

**Decision:**

Accept

**Comment:**

The paper addresses issues in graph contrastive learning that arise in the process of graph augmentation. At the heart of their approach is a simple and intuitive augmentation procedure that "preserves" cohesive topological properties of the initial graph. This may be seen as a limited contribution, but I found the idea interesting and appreciated the theoretical analysis and good experimental results. The authors also made a significant effort to address the reviewer's questions and comments in detail, and I would strongly advise them to incorporate the suggested modifications into their revised manuscript.